# A Novel Curcumin-Mycophenolic Acid Conjugate Inhibited Hyperproliferation of Tumor Necrosis Factor-Alpha-Induced Human Keratinocyte Cells

**DOI:** 10.3390/pharmaceutics13070956

**Published:** 2021-06-25

**Authors:** Yonelian Yuyun, Pahweenvaj Ratnatilaka Na Bhuket, Wiwat Supasena, Piyapan Suwattananuruk, Kemika Praengam, Opa Vajragupta, Chawanphat Muangnoi, Pornchai Rojsitthisak

**Affiliations:** 1Biomedicinal Chemistry Program, Department of Biochemistry and Microbiology, Faculty of Pharmaceutical Sciences, Chulalongkorn University, Bangkok 10330, Thailand; yoneli_redrose@yahoo.com; 2Natural Products for Ageing and Chronic Diseases Research Unit, Chulalongkorn University, Bangkok 10330, Thailand; bpahweenvaj@gmail.com (P.R.N.B.); wiwat.sb@hotmail.com (W.S.); cartoon_ccs@hotmail.com (P.S.); 3Cell and Animal Model Unit, Institute of Nutrition, Mahidol University, Nakhon Pathom 73170, Thailand; kaekemika@gmail.com (K.P.); chawanphat.mua@mahidol.ac.th (C.M.); 4Research Affairs, Faculty of Pharmaceutical Sciences, Chulalongkorn University, Bangkok 10330, Thailand; opa.v@chula.ac.th; 5Department of Food and Pharmaceutical Chemistry, Faculty of Pharmaceutical Sciences, Chulalongkorn University, 254 Phayathai Road, Pathumwan, Bangkok 10330, Thailand

**Keywords:** curcumin, mycophenolic acid, psoriasis, TNF-α induced HaCaT cells, prodrug

## Abstract

Curcumin (CUR) has been used as adjuvant therapy for therapeutic application in the treatment of psoriasis through several mechanisms of action. Due to the poor oral bioavailability of CUR, several approaches have been developed to overcome the limitations of CUR, including the prodrug strategy. In this study, CUR was esterified with mycophenolic acid (MPA) as a novel conjugate prodrug. The MPA-CUR conjugate was structurally elucidated using FT-IR, ^1^H-NMR, ^13^C-NMR, and MS techniques. Bioavailable fractions (BFs) across Caco-2 cells of CUR, MPA, and MPA-CUR were collected for further biological activity evaluation representing an in vitro cellular transport model for oral administration. The antipsoriatic effect of the BFs was determined using antiproliferation and anti-inflammation assays against hyperproliferation of tumor necrosis factor-alpha (TNF-α)-induced human keratinocytes (HaCaT). The BF of MPA-CUR provided better antiproliferation than that of CUR (*p* < 0.001). The enhanced hyperproliferation suppression of the BF of MPA-CUR resulted from the reduction of several inflammatory cytokines, including IL-6, IL-8, and IL-1β. The molecular mechanisms of anti-inflammatory activity were mediated by an attenuated signaling cascade of MAPKs protein, i.e., p38, ERK, and JNK. Our results present evidence for the MPA-CUR conjugate as a promising therapeutic agent for treating psoriasis by antiproliferative and anti-inflammatory actions.

## 1. Introduction

Psoriasis is a skin disease triggered by an environmental or genetic vulnerability that results in an imbalanced immune system characterized by releasing inflammatory cytokines [1]. The clinical manifestations of psoriasis in patients, namely hyperproliferation of keratinocytes, increased neovascularization, and inflammation [1,2]. The exact etiology of psoriasis remains unclear, but nowadays, most treatments involve inhibition of tumor necrosis factor-alpha (TNF-α) production that plays a significant role in skin inflammation [3]. TNF-α is a pro-inflammatory cytokine released mainly by T cells and antigen-presenting cells (APC) in psoriatic skin [4]. TNF-α acts as the central cytokine, regulating the release of other pro-inflammatory cytokines (TNF-α, IFN-γ, IL-6, IL-8, IL-12, IL-17, and IL-18) and chemokines (IL-8 (CXCL8), fractalkine (CX3CL1), CCL5 (RANTES), CCL2 (MCP-1), CCL20 (MIP-3α), CCL26 (eotaxin2), CCL-27 (CTACK), and CXCL10) [4]. TNF-α induced keratinocytes (HaCaT) upregulate intracellular signaling pathways that involve mitogen-activated protein kinases (MAPKs), nuclear factor-kappa B (NF-κB), and caspases [5,6].

Natural products have shown an attractive bioactive source in drug discovery due to their unique structural diversity, leading to many beneficial biological activities [7,8,9]. Many research studies and applications describe their benefits in functional foods, pharmaceuticals, and cosmetics [7]. *Curcuma longa* L. from the Zingiberaceae family is famous for its bioactive compound called curcumin (CUR, Figure 1A) as its main substance. Curcumin has various advantages for human health through its pharmacological activities, such as anticancer, anti-inflammatory, antioxidant, and antiviral properties. The benefits of curcumin have been proven by several in vitro and in vivo experiments [10]. Curcumin also demonstrates antipsoriatic activity through antiproliferation and anti-inflammatory effects. Anti-inflammatory effects of curcumin have been encountered by inhibiting the NF-Kβ, MAPK, and STAT3 pathways, and releasing pro-inflammatory markers, such as IL-17, IL-22, IFN-γ, IL-1β, IL-2, IL-8, IL-6, and TNF-α [1,11,12]. However, curcumin has undesired physicochemical properties, which hinder itself from being developed for clinical applications. For example, instability in alkaline conditions, extensive metabolism, and low oral bioavailability are limitations of curcumin as a therapeutic agent.

Several selective T cell inhibitors are often part of the prescription for psoriasis treatment, including mycophenolic acid (MPA, Figure 1B). MPA works through a mechanism of inhibition of the inosine monophosphate dehydrogenase (IMPDH) enzyme, which is crucial for purine synthesis. Inhibition of IMPDH activity reduces guanosine triphosphate (GTP) and d-guanosine triphosphate (dGTP) levels and drawbacks cell proliferation [13]. In addition, as an anti-inflammatory agent, MPA has demonstrated its ability to reduce the release of pro-inflammatory cytokines such as TNF-α and IL-1β in different cell lines [14,15].

Current psoriasis treatments include conventional therapy (topical, systemic, and phototherapy) and biologics [16]. Vitamin D analogs, dithranol, calcineurin inhibitors, retinoids, coal tar, dithranol, corticosteroids, and emollients are commonly applied as a topical treatment for mild psoriasis. Skin irritation and atrophy are the most reported adverse effect of using a topical application [16,17]. Oral-systemic drugs, such as methotrexate, acitretin and cyclosporine, are prescribed to patients with severe psoriasis. The mechanisms of these drugs are through immunosuppressive and anti-inflammatory activities [18]. Biologics, namely adalimumab, etanercept, and infliximab, are used as TNF-α inhibitors by blocking the role of the cytokine TNF-α [19]. Several clinical studies reported that patients showed no significant response to TNF-α inhibitors, suggesting that there may be a distinction in their inflammatory signaling pathways. The different genetic backgrounds might be the supporting reason for insignificant cure responses [4]. Biologics production spends a high expense, limiting the patient to consume it for the long term because of economic reasons [20,21]. However, TNF-α inhibitors are a good choice for patients diagnosed with moderate to severe psoriasis and psoriatic arthritis who are no longer suitable for conventional treatment [19,22]. Monotherapy usually failed to reach the effectiveness of the treatment and, therefore, combination therapy is preferred to get better efficacy with fewer side effects.

A mutual prodrug is a potential approach for combination therapy because it consists of two bioactive compounds against the same diseases linked by a chemical bond [23]. Traditional combination therapy by combining two drugs in the same formulation may result in the incompatibility of the formulation and the reduction in patient compliance [24]. Applying co-administered medications as a single chemical entity in one dosage form via the mutual prodrug strategy can overcome the above limitations [25]. To date, there have been no reports of using the combined CUR and MPA for psoriasis. For the first time, this study combines a new combination of CUR and MPA using the mutual prodrug approach for psoriasis treatment as an MPA-CUR conjugate (Figure 1C). Curcumin is subjected to various metabolic processes in the liver and intestine, including glucuronidation, sulfation, and reduction after oral administration. Several approaches to enhances the stability and resultant bioavailability of curcumin have been investigated, including the ester prodrug approach [26]. The chemical structure of CUR has three moieties, including two phenolic and one active methylene group, which are available for conjugating with other drugs. Attachment of pro-moieties to phenolic hydroxyl groups of curcumin via biodegradable linkages is common to protect it from degradation [26,27,28]. In terms of the therapeutic effect for psoriasis treatment, CUR acts on many molecular targets in the pathogenesis of psoriasis, such as MAPK suppression and downregulation of several cytokines [1,11,12]. However, MPA has been reported for its adverse effects on the gastrointestinal tract, resulting in dose reduction and, thus, exposing the patient to the risk of treatment failure [29]. The MPA-CUR conjugate may minimize the side effect of MPA if it is effective at a low dose and can overcome metabolic instability of CUR, resulting in an improvement of oral bioavailability.

Our laboratory previously synthesized various CUR prodrugs, for example, curcumin diethyl succinate (CDD) and curcumin diglutaric acid (CurDG). CDD exhibited better stability and cytotoxicity against human colorectal adenocarcinoma (Caco-2) and human hepatocellular carcinoma (HepG2) cells than CUR [25,26,27]. CurDG had improved water solubility, increased oral bioavailability, and enhanced antinociceptive effect compared to CUR [27]. Chopade et al. has reported the prodrug of MPA that contains amino sugars, d-glucosamine (MGLS) and d-galactosamine (MGAS) as pro-moieties. The prodrugs showed increased water solubility, enhanced stability in the upper gastrointestinal homogenate, and decreased ulcer index compared to MPA [30]. Another example, the MPA-triglyceride (TG) mimetic prodrug, showed increased lymphatic transport and mesenteric lymph node concentrations of MPA compared to unmodified MPA [31].

Herein, the MPA-CUR conjugate was synthesized and evaluated for antipsoriatic activity using TNF-α-induced psoriasis-like proliferation of HaCaT cells. The MPA-CUR conjugate was incubated with Caco-2 cells to obtain a bioavailable fraction (BF) of the MPA-CUR conjugate transported across the Caco-2 monolayer for evaluating an antipsoriatic activity. This experimental setting was performed to mimic the absorption and intestinal metabolism of an oral prodrug to produce pharmacological action in vivo. We found that the BF of the MPA-CUR conjugate significantly improved antiproliferative and anti-inflammatory effects against TNF-α-induced HaCaT cells compared to that of CUR.

## 2. Materials and Methods

### 2.1. Materials

MPA (MW 320.3 g/mol) was obtained from AK Scientific (Union City, CA, USA), and CUR (MW 368.4 g/mol) was synthesized and characterized by the previously published method [32]. The 1-Ethyl-3-(3-dimethylamino propyl) carbodiimide hydrochloride (EDC) was obtained from Tokyo Chemical Industry (Tokyo, Japan). Moreover, 4-Dimethylaminopyridine (DMAP) was bought from Sigma Aldrich (St. Louis, MO, USA). HaCaT cell line was purchased from Thermo Fisher Scientific (Waltham, MA, USA). Caco-2 cell line (ATCC No. HTB37) was obtained from the American Type Culture Collection (ATCC, Rockville, MD, USA). The 3-(4,5-dimethylthiazol-2-yl)-2,5-diphenyltetrazolium bromide (MTT) was purchased from Sigma Aldrich (St. Louis, MO, USA). Other materials, including solvents and chemicals, were purchased from commercial sources and used without further purification.

### 2.2. Synthesis of the MPA-CUR Conjugate

A mixture of CUR (368 mg, 1 mmol), MPA (320 mg, 1 mmol), and DMAP (24 mg, 0.2 mmol) was dissolved in acetone (10 mL) to get a clear solution. The mixture was then added dropwise with a solution of EDC (287.55 mg, 1.5 mmol) in acetone (5 mL). The reaction mixture was stirred on an ice bath and kept at the temperature range between 0 and 5 °C for 30 min. To the reaction, 0.1 N HCl (10 mL) was added and stirred for 1 min. The reaction mixture was extracted with dichloromethane (3 × 30 mL). The combined dichloromethane extract was washed with DI water (2 × 20 mL), dried over anhydrous sodium sulfate, and concentrated under reduced pressure (Laborota 4003, Heidolph, Schwabach, Germany) to obtain a crude product in yellow (107 mg, 16% yield). The crude product was subjected to column chromatographic purification on a C18 column (40–63 µ, SiliCycle Inc., Quebec City, Quebec, Canada) using a mixture of methanol and water (8:2) as a mobile phase. Each eluent fraction was monitored by thin-layer chromatography (TLC) on a silica gel 60 F_254_ plate (0.25 mm thickness) using a mixture of dichloromethane and methanol (15:1) as a developing solvent.

### 2.3. Structural Elucidation of MPA-CUR

The synthesized MPA-Cur conjugate was structurally elucidated by attenuated total reflection-Fourier transform infrared spectroscopy (ATR-IR), ^1^H-, and ^13^C-nuclear magnetic resonance spectroscopy (NMR), and high-resolution mass spectrometry (MS). The IR sample was dispersed on the diamond crystal pan. The spectrum was recorded at the wavenumber between 400 and 4000 cm^−1^ using an ATR-FTIR from Bruker INVENIO-S (Billerica, MA, USA). All NMR spectra were recorded on a Bruker 400 Mhz (Billerica, MA, USA) spectrometer, using CDCl_3_ as a solvent and tetramethylsilane (TMS) as an internal standard. Chemical shifts and coupling constants were reported in ppm (δ) and Hz, respectively. The high-resolution mass spectra (HRMS) of all the samples were recorded using a Bruker MicrOTOF-Q II mass spectrometer coupled with an electrospray ion source (Hanau, Germany).

### 2.4. Determination of a Non-Cytotoxic Concentration of MPA-CUR against Caco-2 Cells

Caco-2 cells were cultured in a complete medium culture composed of Dulbecco’s Modified Eagle Medium (DMEM) as a basal medium supplemented with 10% (*v*/*v*) fetal bovine serum (FBS), 1% (*v*/*v*) L-glutamine, 1% (*v*/*v*) nonessential amino acids, 1% (*v*/*v*) penicillin-streptomycin, and 0.2% (*v*/*v*) fungizone. The cells were seeded onto 96-well plates at a density of 1 × 10^4^ cells/200 µL of the complete medium/well. The cells were then incubated at 37 °C in a humidified atmosphere of 95% air: 5% CO_2_ for 24 h. Afterwards, each well was washed and added with 200 µL of the basal medium. The cells were divided into four groups: control, CUR, MPA, and MPA-CUR. DMSO at 0.5% was used as a control. For the treatment group, the compounds were tested at a concentration range of 0.1–10 µM. After a treatment period of 4 h, 20 μL of an MTT solution (in phosphate-buffered saline solution (5 mg/mL) was added to each well and incubated for 4 h. The culture medium was removed before adding DMSO (200 µL) for lysing cells and dissolving formazan crystals. The absorbance of formazan was measured at 540 nm using a microplate reader (CLARIOstar, BMG Labtech, Ortenberg, Germany). Experiments were performed in four replicates and the results were calculated as the percentage of cell viability.

### 2.5. Preparation of Bioavailable Fractions of MPA-CUR

Caco-2 cells were cultured in trans-well inserts of a 6-well plate (ThinCertsTM-TC Einsatze, Greiner Bio-One, St. Gallen, Switzerland) at a density of 2.5 × 10^4^ cells/well in an apical compartment. A 2 mL of phenol red-free DMEM was added to the basolateral compartment. The Caco-2 cells were incubated at 37 °C in a humidified atmosphere of 95% air: 5% CO_2_ for 21–24 days. The confluence of the cultured cells was reached when the concentration of fetal bovine serum (FBS) was decreased to 7.5%. After maintaining the cells for 21–24 days, the monolayer was formed with the transepithelial electrical resistance (TEER) of more than 500 Ω/cm^2^. Each solution of CUR, MPA, or MPA-CUR (2 mL) in serum-free medium was loaded in the apical compartment of the trans-well plate at a final concentration of 5 μM, which is a non-toxic concentration. The treated cells were incubated for 4 h at 37 °C. A blank solution of serum-free medium (2 mL) was performed similarly and the permeate was considered as a blank BF. Experiments were performed in four replicates. The collected BF was blanked with nitrogen gas and stored at −80 °C for in vitro antiproliferation activity assay.

### 2.6. Cell Culture

HaCaT cells were cultured in a complete medium composing of DMEM as a basal medium supplemented with 10% *v*/*v* heat-inactivated FBS and 1% *v*/*v* penicillin-streptomycin at 37 °C in a humidified atmosphere of 5% CO_2_/95% air. All cell lines were seeded in different 96-well plates at a density of 5 × 10^3^ cells/200 µL complete medium/well and incubated at 37 °C in a humidified atmosphere of 5% CO_2_/95% air for 24 h. After seeding for 24 h, the cells were washed with the basal medium followed by the addition of a 200 µL serum-free medium containing 10 ng/mL of TNF-α to induce cell proliferation and inflammation mimicking the psoriatic condition. The treated cells were further incubated for 24 h. Afterwards, the cells were washed with the basal medium.

#### 2.6.1. Direct Treatment

Solutions of CUR, MPA, and MPA-CUR were prepared in 0.5% DMSO at a concentration of 5 μM. 0.5% DMSO was used as the control group. The cells with each sample solution (2 μL) were incubated for 24 h at 37 °C in a humidified atmosphere of 95% air; 5% CO_2_.

#### 2.6.2. BF Treatment

Solutions of the BF of CUR, MPA, or MPA-CUR were individually added to the HaCaT cells. The treated cells were incubated at 37 °C in a humidified atmosphere of 95% air; 5% CO_2_ for 24 h. The blank BF was used as the control group.

#### 2.6.3. Antiproliferation Assay

The proliferation of HaCaT cells under the direct and BF treatment was determined by the MTT assay using the microplate reader for absorbance measurement of formazan. The experiment was performed in four replicates. Results were presented as the percent of cell proliferation. The antiproliferative effects of the tested samples were determined by comparing with the relative number of viable TNF-α-induced HaCaT cells treated with the control group. 0.5% DMSO was used as the control group for the direct treatment group, while the blank BF was used as the control for BF treatment.

### 2.7. Anti-Inflammatory Assay

HaCaT cells were be seeded in 96-well plates. After 24-h incubation, the cells were treated with TNF-α or a combination of TNF-α and BF of CUR or MPA-CUR. After 24 h post-treatment, the culture media were collected for an enzyme-linked immunosorbent assay (ELISA) while protein lysates from cell pellets were prepared for Western blot analysis.

### 2.8. Quantification of IL-6, IL-8, IL-1β by ELISA

Levels of pro-inflammatory cytokines (IL-6, IL-8, and IL-1β) secreted from the TNF-α-induced HaCaT cells were determined using an ELISA kit as described in the manufacturer’s protocols (BioLegend, San Diego, CA, USA). Capture antibodies for IL-6, IL-8, and IL-1β were coated in 96-well plates (NUNC, Roskilde, Denmark). After overnight incubation at 25 °C, the plates were blocked by incubation with 1% bovine serum albumin (BSA) in phosphate-buffered saline for 1 h. Various concentrations of recombinant mouse IL-6, IL-8, and IL-1β solubilized in culture media were used to construct standard curves. The culture media or standard samples were incubated with the coated plates at 25 °C for 2 h before adding biotinylated-detecting antibodies to each well. After 1 h, the immune complex was detected by reacting with a streptavidin-horseradish-HRP-tetramethylbenzidine detection system (Endogen Inc., Rockford, IL, USA). The reaction was stopped by adding 2 M H_2_SO_4_, and the absorbance at 450 nm was immediately determined using the microplate reader. Concentrations of IL-6, IL-8, and IL-1β protein in the culture media samples were calculated by comparing the absorbance with standard curves.

### 2.9. Western Blot Analysis

Cell pellets with equal amounts of protein (40 μg/well) were separated on 8% sodium dodecyl sulfate-polyacrylamide gel electrophoresis (SDS-PAGE) and transferred onto polyvinylidene fluoride membranes (PVDF; Millipore, Bedford, MA, USA). After blocking with 5% non-fat milk in tris-buffered saline (TBST buffer, 0.1% Tween-20) for 1 h at room temperature, the membranes were incubated with primary antibodies for p-JNK, JNK, p-ERK, ERK, p-p38, p38, (1:1000), or β-actin (1:20,000) (Cell Signaling Technology, Danvers, MA, USA) at 4 °C overnight. The membranes were then washed three times with TBST and incubated with horseradish peroxidase-conjugated secondary antibodies for 2 h at room temperature. After visualized using SuperSignal solution (Endogen Inc., Rockford, IL, USA), the immunoblots were exposed to X-Ray films. The protein intensity was measured by the Image J program (freeware downloads from http:/rsb.info.nih.gov/ij (accessed on 25 December 2020)). Results are expressed as a relative ratio of band intensity between proteins. All assays were carried out in triplicate.

### 2.10. Statistical Analysis

All numerical data in this study are presented as the mean ± standard deviation (SD). All the results were analyzed by one-way ANOVA followed by Tukey’s post hoc comparison. Statistical analyses were performed with GraphPad Prism 8 software (San Diego, CA, USA).

## 3. Results and Discussion

### 3.1. Design and Synthesis of MPA-CUR

Although CUR exhibits several pharmacological activities, including antiproliferative and anti-inflammatory effects against TNF-α-induced HaCaT cells [18], rapid metabolism is one of the main reasons for the low oral bioavailability of CUR. Several metabolites were reported when curcumin across Caco-2 cells for in vitro absorption assay [33,34]. The reductive metabolites (hexahydrocurcumin and octahydrocurcumin) and their phase II metabolites (glucuronide and sulfate conjugates) are predominantly found in the Caco-2 cell incubation medium of CUR [33,34]. During intestinal absorption using Caco-2 cell as the in vitro model, CUR is found in only trace amounts because of first-pass metabolism and chemical reduction [33,34]. Therefore, only small amounts of CUR and the conjugates of CUR reach the bloodstream [33]. Regarding MPA, it showed an immunosuppressive effect by inducing the apoptosis of activated T-lymphocytes that leads to anti-inflammation activity [5]. MPA inactivation in vivo is caused mainly by rapid metabolism to 7-O-glucuronide, which is not biologically active. UDP-glucuronosyl transferase act as the catalyst for metabolism activity. These enzymes widely present surrounded by normal tissues, including intestinal epithelium [35,36]. The MPA-CUR prodrug should hinder MPA from glucuronidation to inactive metabolites. According to the metabolism study of MPA in the culture media of human colorectal carcinoma (HT29) by Franklin and colleagues, the chemical modification of phthalane and ringside chains could lead to MPA resistance to rapid conversion. The causes of MPA resistance to metabolism include steric hindrance to substrate accessibility to the glucuronosyltransferase enzyme and the decreased hydrogen bond strength between 7–OH of MPA and adjacent carbonyl functions [36]. Because the MPA-CUR ester prodrug was linked by conjugating an OH–group of CUR with a COOH–group of MPA, CUR, and MPA are considered the potential bioactive molecules. Our experiment used CUR (368.38 mg, 1 mmol) and MPA (320.34 mg, 1 mmol) to get the mutual prodrug of MPA-CUR as a monosubstitution. In this study, the Steglich esterification method, as shown in Figure 2, was used to synthesize the MPA-CUR ester prodrug [37,38].

Our experiment used an equimole of CUR and MPA to obtain the MPA-CUR conjugate as a monosubstitution. The Steglich esterification reaction mediated via the formation of the O-acylisourea derivative of the carboxylic acid in the presence of carbodiimide coupling reagents [37,38]. EDC was used as a carbodiimide coupling reagent while DMAP was used as a Steglich catalyst in the reaction. EDC was selected because the unreacted EDC can be easily removed from the reaction by partitioning with water due to its good water solubility [39]. Several organic solvents, including DCM and DMF (*N*, *N*-dimethylformamide), are commonly used in the Steglich esterification method [37,38,40,41]. Due to the low solubility of curcumin in DCM, a large volume of DCM used to dissolve CUR resulted in a very diluting reaction mixture. For DMF, we have difficulty removing it from the mixture during the workup process owing to the high boiling point of DMF. Therefore, acetone was chosen as a solvent for the coupling reaction due to its low-boiling point. In addition, curcumin can be easily dissolved in acetone and a small volume of acetone is needed. Steglich esterification has been applied to synthesize other MPA or CUR ester prodrugs [30,40,42]. CUR was coupled to several non-steroid anti-inflammatory drugs, namely flufenamic acid, flurbiprofen, naproxen, and ibuprofen with a percentage yield between 17 and 30% [42]. The MPA-CUR conjugate was synthesized successfully, albeit in low yield. It may be attributed to the purification step even though the purification step has been optimized in terms of the mobile and stationary phases. Besides, the TLC result showed that the reaction was not completed after 24 h, similar to the work reported by Laali and colleagues [40]. The MPA-CUR conjugate product was found as yellow solid after purification using C18-bonded silica as the stationary phase and methanol: water (8:2) as the mobile phase. We attempted to use silica gel column chromatography with a combination of hexane and DCM as the mobile phase. Unfortunately, the system failed to separate MPA-CUR from CUR, resulting in the co-elution of MPA-CUR and CUR.

### 3.2. Structural Elucidation of MPA-CUR

The IR spectra of MPA, CUR, and MPA-CUR are shown in Appendix A. The IR analysis confirmed the key functional group of MPA-CUR as it showed the characteristic bands corresponding to the ester bond at 1750.33 cm^−1^(–C=O stretching at C–1), 1267 cm^−1^ (–C–C–O stretching at C–2), and 1124 cm^−1^ (–O–C–C stretching at C–1′). The presence of intramolecular hydrogen bonding between O–21′ and H–O17′ of CUR was marked by the stretching vibration for O–H (O17′-H) group at 3418.46 cm^−1^. The stretching vibration of the conjugated ketone aliphatic at C–9′ was observed at 1627.32 cm^−1^. The C=C (C4′–C5′) stretching vibration of the aromatic ring is observed at 1511.20 cm^−1^. The ^1^H-NMR spectra of MPA-CUR, MPA, and CUR were shown in Appendix A. MPA backbone showed signals for a methyl group at δ 1.79 (H-15, singlet, 3H) and a methyl group linked to the aromatic ring at δ 2.13 (H-17, singlet, 3H), a signal for a methoxyl group at δ 3.74 (H-16, singlet, 3H), two signals corresponding to aliphatic methylenes in the region of δ 2.27 to δ 2.29 (H-3, multiplet, 2H) and δ 2.40 to δ 2.43 (H-2, multiplet, 2H), a broad doublet for a methylene group at δ 3.37 (H-6, 2H), a broad singlet at δ 5.24 (H-10, 2H) referring to an aliphatic methylene bearing oxygen, a multiplet for a methine group of δ 5.22 to δ 5.25 (H-5,1H), and a singlet for an aromatic group bearing hydroxyl δ 7.67 (H-8, 1H). The ^1^H-NMR spectra of MPA-CUR showed both of the methoxy group –OCH_3_-20′ (δ 3.97, s, 3H) and -OCH_3_-21′ (δ 3.86, s, 3H) appeared at the different position, implying that only one phenolic group was esterified to MPA. The ^13^C-NMR spectra of MPA-CUR (Appendix A) supported the infrared data by showing the signals around 171.17 ppm resulting from the carbonyl carbon atom of the ester bond. The experimental ^1^H and ^13^C-NMR chemical shift values for the MPA-CUR conjugate were listed in Table 1. The MPA-CUR conjugate structure was further supported by mass spectral data in Appendix A, in which the [M + Na^+^] peak was observed at *m*/*z* 693.2298 with a mass error <5 ppm.

### 3.3. Antiproliferative Effect of MPA-CUR

Hyperproliferation of keratinocytes is one of the characteristics of psoriasis disease. In our study, we investigated the effect of the MPA-CUR conjugate using hyperproliferative keratinocytes as the in vitro model for psoriasis. Initially, the direct treatment of MPA, CUR and MPA-CUR at the non-toxic concentration of 5 μM against TNF-α-induced HaCaT cells was evaluated for antiproliferative effects. The results showed that the cell viability of MPA-CUR has no significant difference in the cell viability of TNF-α induced HaCaT cells (Figure 3). The results indicated that MPA-CUR has no antiproliferative effect in regards to the indirect treatment. CUR and MPA exhibited the antiproliferative effect compared to MPA-CUR in the direct treatment. As the MPA-CUR conjugate is designed as a prodrug, it is required bioactivation to give active metabolites for exerting pharmacological effects. Consequently, the treatment of a bioavailable fraction (BF) was applied by exposing MPA-CUR to Caco-2 cells for bioconversion during cellular transport. The BF was then used to treat the TNF-α induced HaCaT cells. Prior to that, CUR, MPA, and MPA-CUR were subjected to cytotoxicity assay against Caco-2 cells before cellular transport experiments. The cytotoxic effects of these compounds presented as %cell viability were shown in Figure 4.

The cytotoxicity assay has been evaluated at a range of 0.1–10 µM compared to DMSO 0.5% as the control. MPA did not cause any cytotoxicity in the tested concentration range, in agreement with the observation of Qasim et al. [43]. CUR and MPA-CUR had no significant cytotoxic effect on cell viability in Caco-2 cells at concentrations below 5 µM. Therefore, the concentration of each compound for further experiments was chosen to be 5 μM. Our results are in line with previously reported by Muangnoi et al. using CUR at concentration 5 μM in Caco-2 cells [44]. After incubation of 5 μM of CUR, MPA and MPA-CUR with the Caco-2 cell monolayer for 4 h, the BF in the basolateral compartment were collected and tested on the TNF-α induced HaCaT cells as the model to mimic psoriatic skin. In the TNF-α induced hyperproliferation, TNF-α at 10 ng/mL was suitable for inducing the psoriasis condition by a significant increase in HaCaT cell proliferation by 53% (*p* < 0.001) compared to the control group. This is consistent with our previous report [13]. Then, the antiproliferative effects of the BF from each compound were determined and the results presented as %cell proliferation were shown in Figure 5. The reduction of HaCaT cell proliferation after exposure to the BF of CUR, MPA, and MPA-CUR was 148, 123, and 104% compared to that of the TNF-α control group at 153% (Figure 5). The BF of MPA-CUR more significantly inhibited the hyperproliferation than that of CUR or MPA (*p* < 0.001). We suggested that the MPA-CUR conjugate permeated across the Caco-2 monolayer and released the parent drug into the basolateral compartment to give synergistic antiproliferation effects. The biotransformation of MPA-CUR by esterases in Caco-2 cells might be attributed to the significant effect.

Lau et al. demonstrated that naproxen-dithranol (Nap-DTH), an anti-psoriasis mutual prodrug, had a lower antiproliferative effect than dithranol over 72-h treatment. Furthermore, they tested the Nap-DTH (1:1) mixture, and the antiproliferative effect of the mixture was better than the mutual prodrug itself. Bioconversion Nap-DTH to the parent compounds turned out to have the highest bioactivity [23]. Carboxylesterase (CES), especially human CES1 isoenzyme (hCE1), is the enzyme responsible for the bioconversion of ester prodrugs and has a high expression level in Caco-2 cells [45]. The antiproliferative activity of the BF from MPA-CUR was higher than that of CUR or MPA, possibly due to bioconversion of MPA-CUR to CUR and MPA by CES. The highest percentage of antiproliferation was recorded for the BF of MPA-CUR at 33%, while the direct treatment of MPA-CUR did not exhibit an antiproliferation effect. It means that potential activity from the BF of MPA-CUR resulted from the synergistic effect of the parent drugs that were released in the BF. Direct treatment of CUR denoted an antiproliferative effect at 21%. Meanwhile, in the BF treatment, CUR showed a slight effect at 6% due to its metabolic instability during the transport across Caco-2 cells. Interestingly, for MPA, the BF treatment showed a better antiproliferative effect than the direct treatment, 31% vs. 2%, respectively. Until now, there has been no publication about the active metabolite of MPA from Caco-2 cells. MPA metabolism catalyzed by the enzyme UDP-glucuronosyltransferase (UGT) that presents in the kidneys, liver, and intestine yields two prominent metabolites, namely mycophenolic acid glucuronide (MPAG), an inactive metabolite, and acyl glucuronide (AcMPAG), which is pharmacologically active and found in a lower amount [46]. The molecular mechanisms of MPA-CUR and CUR as the anti-inflammatory agents involved in psoriasis pathogenesis were further investigated by ELISA and Western blot analysis.

### 3.4. MPA-CUR Inhibited the Production of Inflammatory Cytokines

TNF-α has an essential task in managing the production of numerous inflammatory cytokines during psoriasis development in patients. Hence, TNF-α inhibitors have been prescribed for psoriasis patients [16]. In our research, the effect of the BF of MPA-CUR on the level of IL-6, IL-8, and IL-1β was determined by ELISA. Following treatment with TNF-α, cells exhibited enhanced levels of secreted IL-6, IL-8, and IL-1β into the culture medium compared with the control group. As shown in Figure 6, the overexpression of IL-6, IL-8, and IL-1β levels was significantly decreased in the groups treated with the BF of CUR and MPA-CUR at 5 μM. Furthermore, treatment with the BF of MPA-CUR resulted in a significantly pronounced inhibition of IL-6 (59%, *p* < 0.001) and IL-1β (61%, *p* < 0.001) compared to the CUR treatment group (16% and 31% for IL-6 and IL-1β, respectively Figure 6). The inhibitory effects of the BF of MPA-CUR on the IL-8 level were slightly different (52%, *p* = 0.016) compared to that of CUR (28%, Figure 6).

Cytokines released by keratinocytes play a critical role in inflammatory and autoimmune diseases, including psoriasis. Bai et al. revealed a higher amount of several cytokines, including TNF-α, IL-6, and IL-8 in the serum of psoriasis compared to healthy controls [3]. Simultaneously, serum levels of IL-1β were not significantly different between psoriasis patients and controls [3]. As previously published by Sun et al., CUR itself has anti-inflammatory activity by inhibiting TNF-α-induced production of IL-6, IL-8, and IL-1β in HaCaT cells [47]. Our preliminary study (data not shown) demonstrated that the BF of MPA did not show an anti-inflammatory effect in the TNF-α-induced HaCaT cell model. These findings are in line with the studies by Kim and colleagues (2015) [5]. Regarding the loss of the cytokine inhibitory effect of MPA, we hypothesized that the MPA concentration in BF is too low to demonstrate cytokine inhibition. Baer et al. (2004) reported that MPA did not show the inhibitory effect on the IL-6 level compared to the IL-1β stimulation group in human proximal cells (PTC) and distal tubular cells (DTC) at a low dose 0.25 μM [48]. The ability of the BF of MPA-CUR in reducing cytokine production suggested the possible use of MPA-CUR for psoriasis treatment and the inflammatory signaling pathways were further investigated.

### 3.5. MPA-CUR Inhibited the p38, ERK, and JNK Phosphorylation

TNF-α is a cytokine that activates various signaling pathways, including MAPK, resulting in keratinocyte inflammation. To further determine the mechanism underlying the anti-inflammatory effects of the BF of MPA-CUR, we evaluated the role of MAPK signaling cascades in TNF-α-inflammatory responses in HaCaT cells. TNF-α exposure significantly increased p38 (*p* < 0.001), ERK (*p* < 0.001), and JNK (*p* < 0.001) phosphorylation in HaCaT cells after 24-h treatment followed by 24-h incubation compared to the untreated cells. The BF of MPA-CUR significantly reduced (*p* < 0.001) the TNF-α induced phosphorylation of all three proteins, i.e., p38, ERK and JNK (50, 41, and 48%, respectively), significantly greater (*p* < 0.001) than that of the BF of CUR (Figure 7).

## 4. Conclusions

A mutual prodrug of MPA and CUR for the treatment of psoriasis has been synthesized using the Steglich esterification between MPA and CUR. The MPA-CUR conjugate exhibited an improved antiproliferation effect against TNF-α induced HaCaT cells. The MPA-CUR conjugate was proposed as a mutual prodrug that the parent compounds showed synergistic action in the bioavailable fraction from Caco-2 cells. MPA-CUR inhibited inflammatory cytokines, such as IL-6, IL-8, and IL-1β in TNF-α induced HaCaT cells through the attenuation of the MAPK signaling pathways, including p38, ERK, and JNK. These findings demonstrated the antiproliferative and anti-inflammatory effects of MPA-CUR that could be further developed for the treatment of psoriasis. It is worth noting that the MPA-CUR conjugate synthesized in this study will be considered as a novel chemical entity and will not be a product in the food list as curcumin [49]. To develop the conjugate as a potential drug candidate, further studies, such as the investigation of physicochemical properties, biopharmaceutics classification, animal pharmacokinetics, and in vivo antipsoriatic activities, should be performed. Imiquimod (IMQ)-induced wild-type mouse mimicking some features of psoriasis can be used as an animal model for an antipsoriatic assay for the conjugate.

## Figures and Tables

**Figure 1 pharmaceutics-13-00956-f001:**
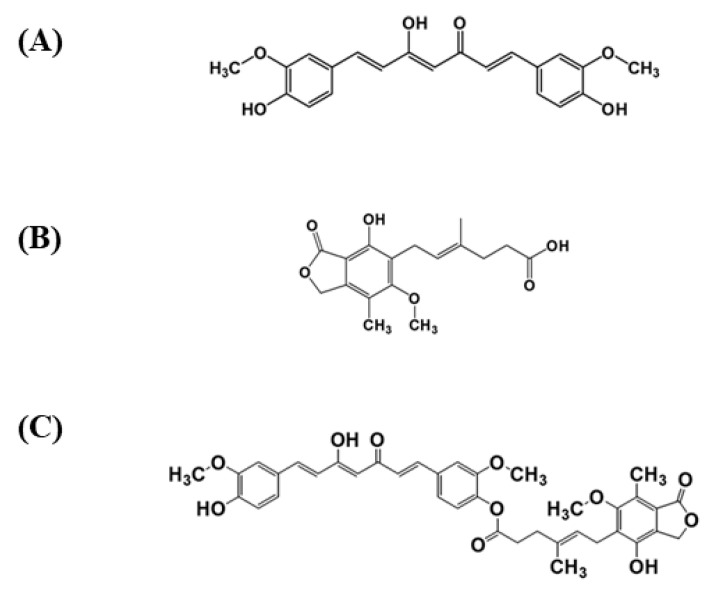
Chemical structures of (**A**) curcumin, CUR (**B**) mycophenolic acid, MPA, and (**C**) mycophenolic acid-curcumin, MPA-CUR.

**Figure 2 pharmaceutics-13-00956-f002:**
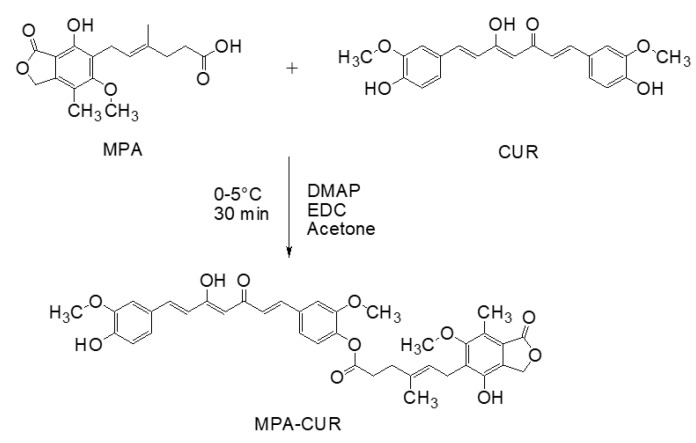
Synthesis of the MPA-CUR conjugate.

**Figure 3 pharmaceutics-13-00956-f003:**
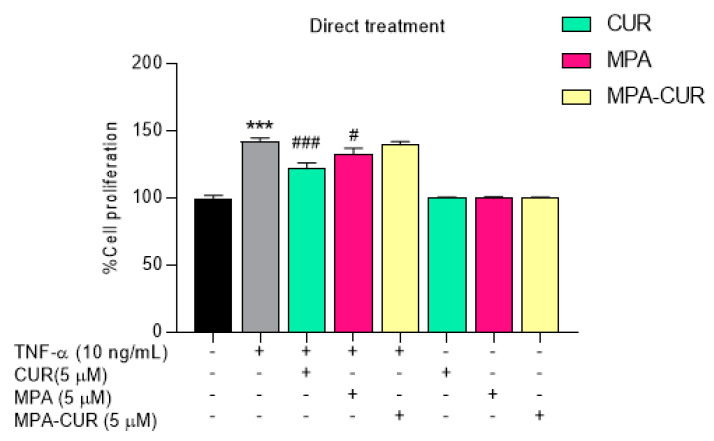
Cell proliferation of TNF-α-induced HaCaT cells incubated with MPA, CUR, and MPA-CUR for 24 h in the direct treatment. Data presented are mean ± SD values of three replicates *** *p* < 0.001 indicates significance from the untreated control cells, ^#^ *p* < 0.05 and ^###^ *p* < 0.001 indicates significance from TNF-α control group.

**Figure 4 pharmaceutics-13-00956-f004:**
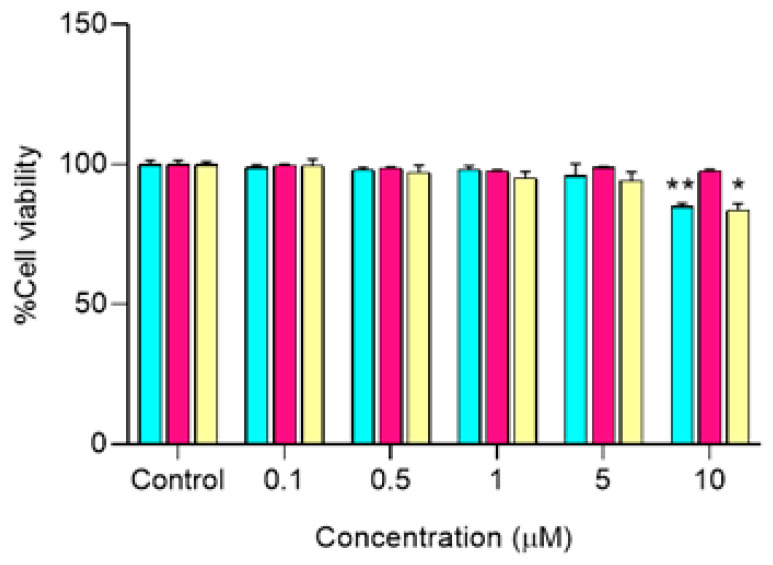
Cell viability of Caco-2 cells incubated with CUR, MPA, and MPA-CUR over a concentration range of 0.1–10 μM for 24 h. Data presented are mean ± SD values of four replicates * *p* < 0.05 and ** *p* < 0.01 indicates significance from the control group.

**Figure 5 pharmaceutics-13-00956-f005:**
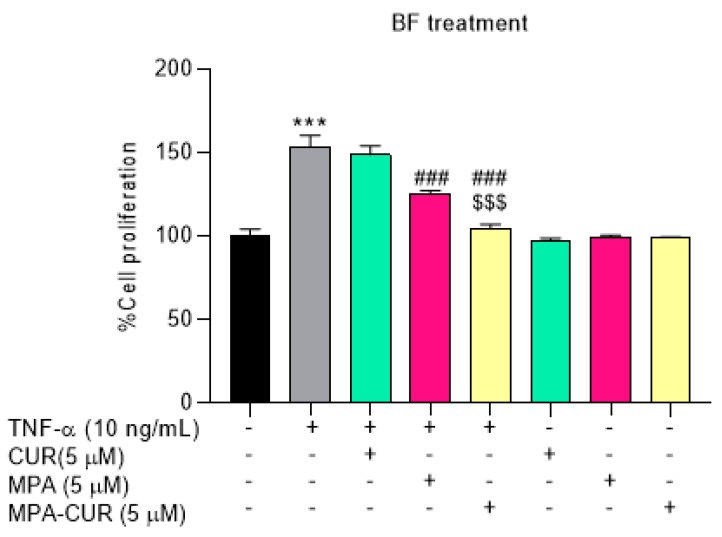
Cell proliferation of TNF-α-induced HaCaT cells incubated with MPA, CUR, and MPA-CUR for 24 h in BF treatment. Data presented are mean ± SD values of three replicates *** *p* < 0.001 indicates significance from the untreated control cells, ^###^ *p* < 0.001 indicates significance from TNF-α control group and ^$$$^ *p* < 0.001 indicates significance from CUR and MPA treated groups.

**Figure 6 pharmaceutics-13-00956-f006:**
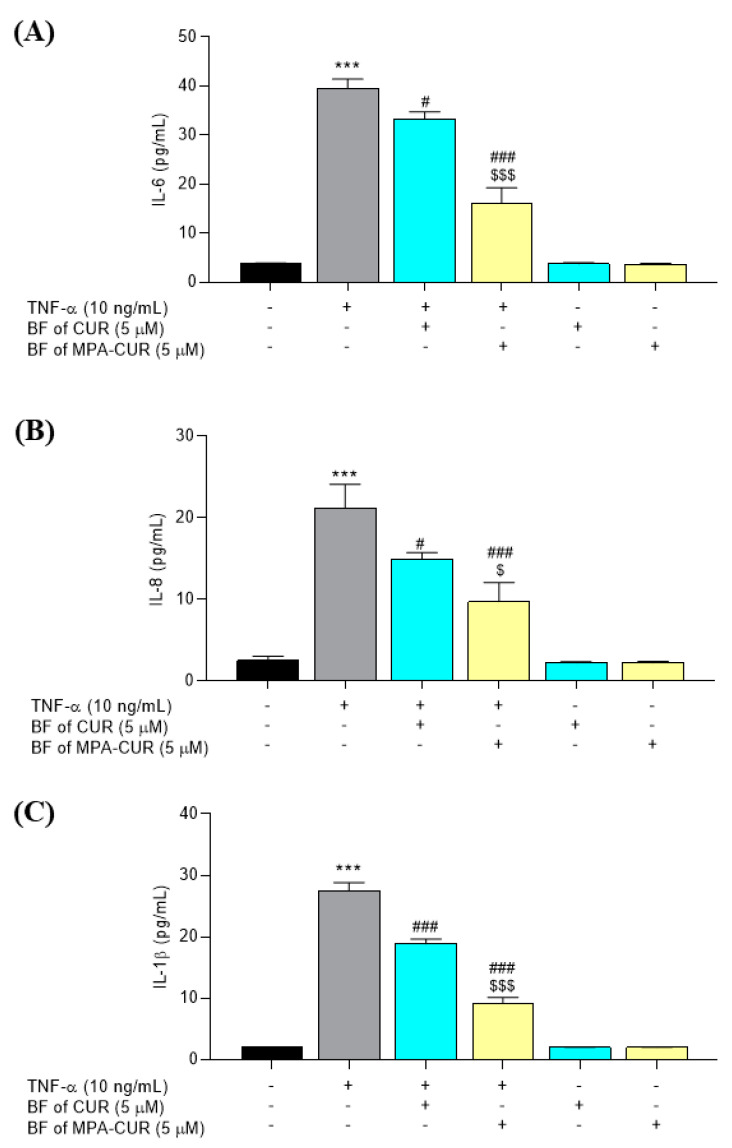
The effects of the BF of MPA-CUR on the expression of inflammatory cytokines, IL-6, IL-8, and IL-1β, following treatment with TNF-α in HaCaT cells (**A**–**C**). Data presented are mean ± SD values of three replicates *** *p* < 0.001 indicates significance from the untreated control cells, ^#^ *p* < 0.05 and ^###^ *p* < 0.001 indicates significance from the TNF-α control group and ^$^ *p* < 0.05 and ^$$$^ *p* < 0.001 indicates significance from the BF of CUR treated group.

**Figure 7 pharmaceutics-13-00956-f007:**
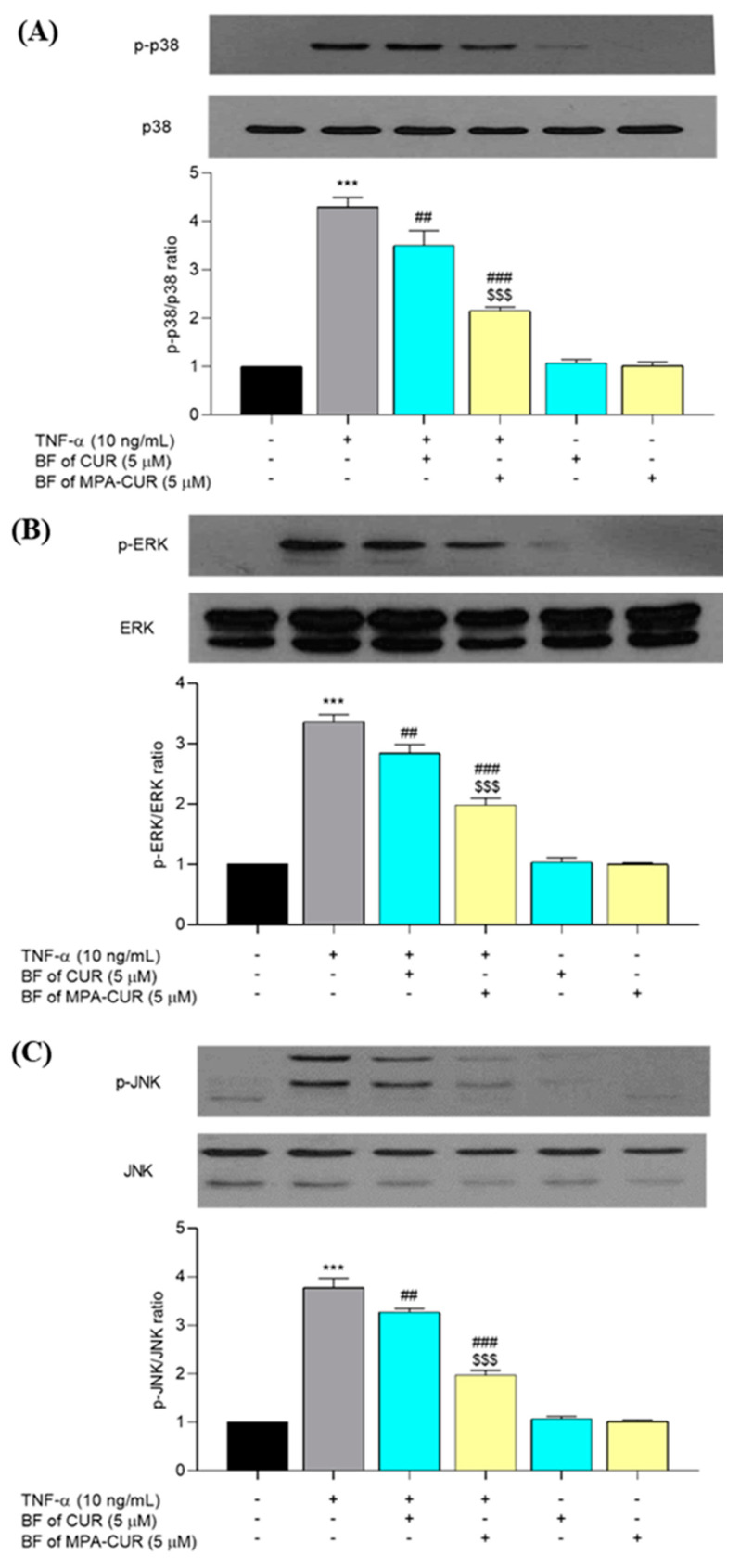
The effects of the BF of MPA-CUR on MAPK signaling pathway, following treatment with TNF-α in HaCaT cells. The protein expression levels of p-p38, p38, p-ERK, ERK, p-JNK, and JNK in HaCaT cells were detected by Western blot analysis (**A**–**C**). Data presented are mean ± SD values of three replicates *** *p* < 0.001 indicates significance from the untreated control cells, ^##^ *p* < 0.01 and ^###^ *p* < 0.001 indicates significance from the TNF-α control group and ^$$$^ *p* < 0.001 indicates significance from BF of CUR treated group.

**Table 1 pharmaceutics-13-00956-t001:** ^1^H and ^13^C NMR spectral data of MPA-CUR conjugate in chloroform-*d*.

Position	^1^H-NMR (ppm)	^13^C-NMR (ppm)
1, 10	-	171.17, 172.97
2, 3	2.70 (t, *J* = 7.6 Hz), 2.47 (t, *J* = 7.6 Hz)	32.77, 34.59
4, 7,12, 4′, 14′	-	127.58, 123.07, 144.07, 139.42, 124.22
5	5.36 (t, *J* = 6.6 Hz)	122.1
6	3.45 (d, *J* = 6.9 Hz)	22.66
8, 14, 17′, 18′	-	153.65, 163.72, 141.25, 146.83
9, 13	-	106.39, 116.77
11, 16	5.22 (s), 3.79 (s)	70.07, 61.04
15, 17	1.89 (s), 2.17 (s)	16.17, 11.6
1′, 2′	-	141.11, 151.41
3′, 5′	7.15 (m), 7.10 (m)	109.68, 114.87
Position	1H-NMR (ppm)	13C-NMR (in ppm)
6′, 16′	6.96 (t, *J* = 16.6 Hz), 6.99 (t, *J* = 16.6 Hz)	123.07, 111.42
7′	7.62 (dd, *J* = 15.9 Hz)	133.95
8′	6.58 (t, *J* = 16.6 Hz)	120.89
9′	-	184.48
10′	5.85 (s)	101.54
11′	-	181.87
12′	6.53 (t, *J* = 16.6 Hz)	123.07
13′	7.56 (dd, *J* = 15.8 Hz)	148.02
15′, 19′	7.07 (m), 7.13 (m)	121.8, 109.68
20′, 21′	3.97 (s), 3.86 (s)	55.99, 55.91
Enolic proton	7.72 (s)	-

## Data Availability

Not applicable.

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
