# Peer review of "A Novel Curcumin-Mycophenolic Acid Conjugate Inhibited Hyperproliferation of Tumor Necrosis Factor-Alpha-Induced Human Keratinocyte Cells"

_pharmaceutics, 2021, doi:10.3390/pharmaceutics13070956_

Round 1
Reviewer 1 Report
The manuscript "A Novel Curcumin-Mycophenolic acid Conjugate Inhibited Hyperproliferation of Tumor Necrosis Factor-Alpha-Induced Human Keratinocyte Cells" submitted to Pharmaceutics for publication descirbes the formulation of a conjugate for the use in psoriasis. I think that the idea is original but not well explained in the design section of the introduction. Please explain why these two agents were selected.
The methods are clearly described, but some questions are:
why did you use acetone for the coupling reaction, DMF or DCM could not be used?
The events are not clear in the synthetic approach.
The results section is well written and described , but I have a curiosity:
it could be useful to perform an assay in presence of corticosteroid to better validate the obtained conjugate.
Author Response
Comments from Reviewer #1
The manuscript "A Novel Curcumin-Mycophenolic Acid Conjugate Inhibited Hyperproliferation of Tumor Necrosis Factor-Alpha-Induced Human Keratinocyte Cells" submitted to Pharmaceutics for publication describes the formulation of a conjugate for the use in psoriasis. I think that the idea is original but not well explained in the design section of the introduction. Please explain why these two agents were selected. The methods are clearly described, but some questions are: Why did you use acetone for the coupling reaction, DMF or DCM could not be used? The events are not clear in the synthetic approach. The results section is well written and described, but I have curiosity: it could be useful to perform an assay in presence of corticosteroid to better validate the obtained conjugate.
Responses to the comments
Point#1: Please explain why these two agents were selected.
Response:
Thank you very much for your comment. Curcumin is subjected to various metabolic processes in the liver and intestine including glucuronidation, sulfation, and reduction after oral administration. Several approaches to enhances the stability and resultant bioavailability of curcumin have been investigated, including the ester prodrug approach. The chemical structure of CUR has three moieties, including two phenolic and one active methylene group, that are available for conjugating with other drugs. Attachment of pro-moieties to phenolic hydroxyl groups of curcumin via biodegradable linkages is common to protect it from degradation [26-28].
In terms of the therapeutic effect for psoriasis treatment, CUR acts on many molecular targets in the pathogenesis of psoriases such as MAPK suppression and down-regulation of several cytokines [1, 11, 12]. CUR could therefore be enhanced through a synergistic effect of MPA as an antiproliferative agent. However, MPA has been reported for its adverse effects on the gastrointestinal tract, resulting in dose reduction and, thus, exposing the patient to the risk of treatment failure [27]. The MPA-CUR conjugate may minimize the side effect of MPA if it is effective at a low dose and can overcome metabolic instability of CUR, resulting in an improvement of oral bioavailability.
We have added the above information in the introduction section of the revised manuscript as indicated with the Track Change Function. The added sentences are also shown below.
Added sentences in the revised manuscript (Page 3, Line 118-122):
Curcumin is subjected to various metabolic processes in the liver and intestine including glucuronidation, sulfation, and reduction after oral administration. Several approaches to enhances the stability and resultant bioavailability of curcumin have been investigated, including the ester prodrug approach [26].
Added sentences in the revised manuscript (Page 3, Line 124-125):
Attachment of pro-moieties to phenolic hydroxyl groups of curcumin via biodegradable linkages is common to protect it from degradation [26-28].
Added sentences in the revised manuscript (Page 3, Line 129-130):
In terms of the therapeutic effect for psoriasis treatment, CUR acts on many molecular targets in the pathogenesis of psoriases such as MAPK suppression and down-regulation of several cytokines [1, 11, 12]. CUR could therefore be enhanced through a synergistic effect of MPA as an antiproliferative agent. However, MPA has been reported for its adverse effects on the gastrointestinal tract, resulting in dose reduction and, thus, exposing the patient to the risk of treatment failure [29]. The MPA-CUR conjugate may minimize the side effect of MPA if it is effective at a low dose and can overcome metabolic instability of CUR, resulting in an improvement of oral bioavailability.
References:
- Varma, S.R., et al., Imiquimod-induced psoriasis-like inflammation in differentiated Human keratinocytes: Its evaluation using curcumin. Eur J Pharmacol, 2017. 813: p. 33-41.
- Kang, D., et al., Curcumin shows excellent therapeutic effect on psoriasis in mouse model. Biochimie, 2016. 123: p. 73-80.
- Cho, J.W., K.S. Lee, and C.W. Kim, Curcumin attenuates the expression of IL-1beta, IL-6, and TNF-alpha as well as cyclin E in TNF-alpha-treated HaCaT cells; NF-kappaB and MAPKs as potential upstream targets. Int J Mol Med, 2007. 19(3): p. 469-74.
- Ratnatilaka Na Bhuket, P., et al., Enhancement of Curcumin Bioavailability Via the Prodrug Approach: Challenges and Prospects. Eur J Drug Metab Pharmacokinet, 2017. 42(3): p. 341-353.
- Muangnoi, C., et al., A curcumin-diglutaric acid conjugated prodrug with improved water solubility and antinociceptive properties compared to curcumin. Biosci Biotechnol Biochem, 2018. 82(8): p. 1301-1308.
- Wichitnithad, W., et al., Synthesis, characterization and biological evaluation of succinate prodrugs of curcuminoids for colon cancer treatment. Molecules, 2011. 16(2): p. 1888-1900.
- Davies, N.M., et al., Gastrointestinal side effects of mycophenolic acid in renal transplant patients: a reappraisal. Nephrol Dial Transplant, 2007. 22(9): p. 2440-8.
Point#2: Why did you use acetone for the coupling reaction, DMF or DCM could not be used?
Response:
Thank you very much for the comments. In this work, several organic solvents including dichloromethane (DCM), N, N-dimethylformamide (DMF) and acetone. Due to the low solubility of curcumin in DCM, a large volume of DCM was needed resulting in a very diluting reaction mixture. For DMF, we have difficulty removing it from the mixture during the workup process due to the high boiling point of DMF. Since acetone has a low-boiling point and can dissolve curcumin easily, it is chosen in this study. We have discussed this issue in the discussion section (page 7, lines 343-350).
Added Sentences (page 7, lines 343-350):
Several organic solvents, including DCM and DMF (N, N-dimethylformamide), are commonly used in the Steglich esterification method [37, 38, 40, 41]. Due to the low solubility of curcumin in DCM, a large volume of DCM used to dissolve CUR resulted in a very diluting reaction mixture. For DMF, we have difficulty removing it from the mixture during the workup process owing to the high boiling point of DMF. Therefore, acetone was chosen as a solvent for the coupling reaction due to its low-boiling point. In addition, curcumin can be easily dissolved in acetone and a small volume of acetone is needed.
Point#3: The events are not clear in the synthetic approach.
Response:
Thank you for the comment. The questions have been addressed in the methodology and discussion sections with appropriate references as indicated with the Track Change Function in the revised manuscript
Added sentences in the methodology section (page 4, lines 165-178):
A mixture of CUR (368 mg, 1 mmol), MPA (320 mg, 1 mmol) and DMAP (24 mg, 0.2 mmol) was dissolved in acetone (10 mL) to get a clear solution. The mixture was then added dropwise with a solution of EDC (287.55 mg, 1.5 mmol) in acetone (5 mL). The reaction mixture was stirred on an ice bath and kept at the temperature range between 0-5°C for 30 min. To the reaction, 0.1 N HCl (10 mL) was added and stirred for 1 min. The reaction mixture was extracted with dichloromethane (3 x 30 mL). The combined dichloromethane extract was washed with DI water (2 x 20 mL), dried over anhydrous sodium sulfate, and concentrated under reduced pressure (Laborota 4003, Heidolph, Schwabach, Germany) to obtain a crude product in yellow (107 mg, 16% yield). The crude product was subjected to column chromatographic purification on a C18 column (40-63 µ, SiliCycle Inc., Quebec, Canada) using a mixture of methanol and water (8:2) as a mobile phase. Each eluent fraction was monitored by thin-layer chromatography (TLC) on a silica gel 60 F254 plate (0.25 mm thickness) using a mixture of dichloromethane and methanol (15:1) as a developing solvent.
Added sentences in the discussion section (page 7, lines 327-366):
In this study, the Steglich esterification method as shown in Figure 2. was used to synthesize the CUR-MPA ester prodrug [37, 38]. Our experiment used an equimole of CUR and MPA to obtain the MPA-CUR conjugate as a monosubstitution. The Steglich esterification reaction mediated via the formation of the O-acylisourea derivative of the carboxylic acid in the presence of carbodiimide coupling reagents [37, 38]. EDC was used as a carbodiimide coupling reagent while DMAP was used as a Steglich catalyst in the reaction. EDC was selected because the unreacted EDC can be easily removed from the reaction by partitioning with water due to its good water solubility [39]. Several organic solvents, including DCM and DMF (N, N-dimethylformamide), are commonly used in the Steglich esterification method [37, 38, 40, 41]. Due to the low solubility of curcumin in DCM, a large volume of DCM used to dissolve CUR resulted in a very diluting reaction mixture. For DMF, we have difficulty removing it from the mixture during the workup process owing to the high boiling point of DMF. Therefore, acetone was chosen as a solvent for the coupling reaction due to its low-boiling point. In addition, curcumin can be easily dissolved in acetone and a small volume of acetone is needed.
Point#4: The results section is well written and described, but I have a curiosity: it could be useful to perform an assay in presence of corticosteroid to better validate the obtained conjugate.
Response:
Thank you for bringing up this idea. Our study aimed to enhance the antipsoriatic activity of CUR and MPA with the MPA-CUR conjugate using the TNF-a-induced HaCaT cell model. Therefore, CUR or MPA could be used as an indirect positive control because they were demonstrated to possess anti-psoriasis effects [1-2]. To the best of our knowledge, there was no report of using corticosteroids as the positive control [3-7]. However, the TNF-α induced HaCaT cell proliferation by approximately 50% compared to the control group in the absence of the inducer has been considered as an indicator of the suitability of the model for each experiment [5,7].
References:
- Supasena, W., et al., Enhanced Antipsoriatic Activity of Mycophenolic Acid Against the TNF-alpha-Induced HaCaT Cell Proliferation by Conjugated Poloxamer Micelles. J Pharm Sci, 2020. 109(2): p. 1153-1160.
- Cho, J.W., K.S. Lee, and C.W. Kim, Curcumin attenuates the expression of IL-1beta, IL-6, and TNF-alpha as well as cyclin E in TNF-alpha-treated HaCaT cells; NF-kappaB and MAPKs as potential upstream targets. Int J Mol Med, 2007. 19(3): p. 469-74.
- Xiong H, Xu Y, Tan G, Han Y, Tang Z, Xu W, Zeng F, Guo Q: Glycyrrhizin Ameliorates Imiquimod-Induced Psoriasis-like Skin Lesions in BALB/c Mice and Inhibits TNF-a-Induced ICAM-1 Expression via NFkB/MAPK in HaCaT Cells. Cell Physiol Biochem 2015;35:1335-1346. doi: 10.1159/000373955.
- Aimin Liu, Wei Zhao, Buxin Zhang, Yuanhui Tu, Qingxing Wang, Jing Li; Cimifugin ameliorates imiquimod-induced psoriasis by inhibiting oxidative stress and inflammation via NF-κB/MAPK pathway. Biosci Rep 26 June 2020; 40 (6): BSR20200471. doi: https://doi.org/10.1042/BSR20200471.
- Leng, H., Pu, L., Xu, L., Shi, X., Ji, J., & Chen, K. (2018). Effects of aloe polysaccharide, a polysaccharide extracted from Aloe vera, on TNF‑α‑induced HaCaT cell proliferation and the underlying mechanism in psoriasis. Molecular Medicine Reports, 18, 3537-3543. https://doi.org/10.3892/mmr.2018.9319.
- Lu Yue, Wang Ailin, Zhang Jinwei, Li Leng, Wei Jianan, Li Li, Chen Haiming, Han Ling, Lu Chuanjian, PSORI-CM02 ameliorates psoriasis in vivo and in vitro by inducing autophagy via inhibition of the PI3K/Akt/mTOR pathway, Phytomedicine, Volume 64, 2019, 153054, ISSN 0944-7113, https://doi.org/10.1016/j.phymed.2019.153054.
- Wu, X., Deng, X., Wang, J., & Li, Q. (2020). Baicalin Inhibits Cell Proliferation and Inflammatory Cytokines Induced by Tumor Necrosis Factor α (TNF-α) in Human Immortalized Keratinocytes (HaCaT) Human Keratinocytes by Inhibiting the STAT3/Nuclear Factor kappa B (NF-κB) Signaling Pathway. Medical science monitor: international medical journal of experimental and clinical research, 26, e919392. https://doi.org/10.12659/MSM.919392

Reviewer 2 Report
The paper entitled: "A Novel Curcumin-Mycophenolic acid Conjugate Inhibited 2 Hyperproliferation of Tumor Necrosis Factor-Alpha-Induced 3 Human Keratinocyte Cells", is a study that explores the effect of curcumin (CUR) esterified with mycophenolic acid (MPA), and named as the MPA-CUR conjugate, to be used as a therapeutic option on psoriasis.
The structure of this conjugate and its metabolites or bioavailable fractions (BFs) were analyzed. In the latter, the viability of Caco-2 and HaCaT cells in which hyperproliferation was induced by the addition of TNF-α was sought.
MPA-CUR BF was found to have better antiproliferative effect than CUR alone. This was observed with the decrease of the cytokines IL-6, IL-8 and IL-1β, due to an attenuated signaling of the MAPK cascade.
Author Response
Comments from Reviewer #2
The paper entitled: "A Novel Curcumin-Mycophenolic acid Conjugate Inhibited 2 Hyperproliferation of Tumor Necrosis Factor-Alpha-Induced Human Keratinocyte Cells", is a study that explores the effect of curcumin (CUR) esterified with mycophenolic acid (MPA), and named as the MPA-CUR conjugate, to be used as a therapeutic option on psoriasis. The structure of this conjugate and its metabolites or bioavailable fractions (BFs) were analyzed. In the latter, the viability of Caco-2 and HaCaT cells in which hyperproliferation was induced by the addition of TNF-α was sought. MPA-CUR BF was found to have better antiproliferative effect than CUR alone. This was observed with the decrease of the cytokines IL-6, IL-8 and IL-1β, due to an attenuated signaling of the MAPK cascade.
Responses to the comments
We are grateful to the reviewer for the positive and encouraging comments.

Reviewer 3 Report
The manuscript “A Novel Curcumin-Mycophenolic acid Conjugate Inhibited Hyperproliferation of Tumour Necrosis Factor-Alpha-Induced Human Keratinocyte Cells” by Yuyun et al is very well written with robust experimental design and established rational. The products are very well characterised and assessed.
Have authors assessed the solubility of their conjugates?
Is there any release/cleavage study of the drug form the prodrug
Have the authors consider a stability study of the prodrugs on different media?
Chemical modification of Curcumin will remove the product form food list and add to chemical list- different regulations – is the conjugation worth it? What is the next step?
Author Response
Comments from Reviewer #3
The manuscript “A Novel Curcumin-Mycophenolic acid Conjugate Inhibited Hyperproliferation of Tumour Necrosis Factor-Alpha-Induced Human Keratinocyte Cells” by Yuyun et al is very well written with robust experimental design and established rational. The products are very well characterised and assessed.
- Have authors assessed the solubility of their conjugates?
- Is there any release/cleavage study of the drug from the prodrug? Have the authors consider a stability study of the prodrugs on different media?
- Chemical modification of Curcumin will remove the product from food list and add to chemical list- different regulations – is the conjugation worth it? What is the next step?
Responses to the comments
Point#1: Have authors assessed the solubility of their conjugates?
Response:
Thank you very much for your suggestion. We have not yet performed the solubility of the conjugate, and therefore we have not had results to include in this present manuscript. We plan to investigate the physicochemical properties including water solubility, stability and log P of the conjugates in future research.
Point#2: Is there any release/cleavage study of the drug from the prodrug? Have the authors consider a stability study of the prodrugs on different media?
Response:
Thank you very much for your suggestion. We have not yet performed the release study of the conjugate, and therefore we have not had results to include in this present manuscript. We plan to investigate the physicochemical properties including water solubility, stability and log P of the conjugates in future research.
Point#3: Chemical modification of Curcumin will remove the product from food list and add to chemical list- different regulations – is the conjugation worth it? What is the next step?
Response:
Thank you very much for raising this concern. The current study aims to improve the anti psoriasis activity of curcumin for therapeutic purposes. Therefore, the conjugate is considered to be a novel chemical entity that can hopefully be further developed as a potential drug candidate. Our further studies on the CUR-MPA conjugate included the investigation of physicochemical properties, biopharmaceutics classification, animal pharmacokinetics and in vivo antipsoriasis activities. We have addressed your suggestion and concern in the conclusion section of the revised manuscript (page 15, lines 675-682).
Added sentences in the conclusion section (page 15, lines 675-682):
It is worth noting that the CUR-MPA conjugate synthesized in this study will be considered as a novel chemical entity and will not be a product in the food list as curcumin [49]. To develop the conjugate as a potential drug candidate, further studies such as the investigation of physicochemical properties, biopharmaceutics classification, animal pharmacokinetics and in vivo antipsoriasis activities should be performed. Imiquimod (IMQ) induced wild-type mouse to mimic some features of psoriasis can be used as an animal model for an antipsoriasis assay for the conjugate.
Reference:
- Hewlings, S. J., & Kalman, D. S. (2017). Curcumin: A Review of Its Effects on Human Health. Foods (Basel, Switzerland), 6(10), 92. https://doi.org/10.3390/foods6100092

Reviewer 4 Report
The study investigated therapeutic effects of curcumin as an anti-inflammatory agent for psoriasis symptoms. The study is interesting and addressed immune responses relevant to psoriasis. As a good sign of novelty, authors developed and used curcumin conjugates that were not previously tested as immunomodulators. However, there are several problems to amend.
- The paper requires minor English academic editing to clarify several phrases. There are several long phrases that should be separated into 2- smaller sentences.
- The first sentence in the abstract contains unclear part: “Curcumin (CUR) has been used as adjuvant therapy for therapeutic application in the treatment of psoriasis by several mechanisms”. Several mechanisms of treatment ( methods?) or several mechanisms of curcumin signaling in cells? This sentence should be re-phrase.
- Line 48: “Natural products have shown an attractive bioactive source in drug discovery due to their unique structural diversity [7].”. The structural diversity is not the main beneficial characteristics. The biological effect of the natural compounds is the reason why the agents are used for drug discovery. It is necessary to cite relevant references here.
- Fig.6 ; western blots/cuts are too small for clear visualization; and should be increased in size. The size of the graphs can be decreased.
- Study limitations should be described. Future directions ( testing in vivo, which models ?…)should be indicated.
- Conclusions are too general. Avoid vacuous generalizations. Conclusions should indicate the findings ( cytokines, proliferation-related effects etc)
Author Response
Comments from Reviewer #4
The study investigated therapeutic effects of curcumin as an anti-inflammatory agent for psoriasis symptoms. The study is interesting and addressed immune responses relevant to psoriasis. As a good sign of novelty, authors developed and used curcumin conjugates that were not previously tested as immunomodulators. However, there are several problems to amend.
- The paper requires minor English academic editing to clarify several phrases. There are several long phrases that should be separated into 2- smaller sentences.
- The first sentence in the abstract contains unclear part:“Curcumin (CUR) has been used as adjuvant therapy for therapeutic application in the treatment of psoriasis by several mechanisms”. Several mechanisms of treatment (methods?) or several mechanisms of curcumin signaling in cells? This sentence should be re-phrased.
- Line 48: “Natural products have shown an attractive bioactive source in drug discovery due to their unique structural diversity [7].” The structural diversity is not the main beneficial characteristics. The biological effect of the natural compounds is the reason why the agents are used for drug discovery. It is necessary to cite relevant references here.
- Fig. 6; western blots/cuts are too small for clear visualization and should be increased in size. The size of the graphs can be decreased.
- Study limitations should be described. Future directions (testing in vivo, which models ?…) should be indicated.
- Conclusions are too general. Avoid vacuous generalizations. Conclusions should indicate the findings ( cytokines,proliferation-related effects etc.)
Responses to the comments
Point#1: The paper requires minor English academic editing to clarify several phrases. There are several long phrases that should be separated into 2- smaller sentences.
Response:
Thank you very much for your comments. We have carefully edited several long sentences and split them into shorter sentences. The details are shown below.
- Original sentence (page 2, lines 95-98):
Oral-systemic drugs like methotrexate, acitretin and cyclosporine are prescribed to patients with severe psoriasis in which the mechanisms of these drugs are through immunosuppressive and anti-inflammatory activities.
Revised sentences (page 2, lines 98-100):
Oral-systemic drugs such as methotrexate, acitretin, and cyclosporine are prescribed to patients with severe psoriasis. The mechanisms of these drugs are through immunosuppressive and anti-inflammatory activities.
- Original sentence (page 3, lines 125-129):
The therapeutic effect for psoriasis treatment would be enhanced by a synergistic effect of MPA as an antiproliferative agent and CUR that acts on many molecular targets in the pathogenesis of psoriasis such as MAPK suppression and down-regulation of several cytokines.
Revised sentences (page 3, lines 129-131):
In terms of the therapeutic effect for psoriasis treatment, CUR acts on many molecular targets in the pathogenesis of psoriases such as MAPK suppression and down-regulation of several cytokines.
- Original sentence (page 3, 131-133):
MPA has been reported for its adverse reaction to the gastrointestinal tract, resulting in dose reduction and, thus, exposing the patient to the risk of treatment failure.
Revised sentences (page 3, lines 133-135):
However, MPA has been reported for its adverse effects on the gastrointestinal tract, resulting in dose reduction and, thus, exposing the patient to the risk of treatment failure.
- Original sentence (page 3, lines 135-137):
MPA-CUR conjugate can minimize the side effects of MPA due to using low doses. In addition, MPA-CUR conjugates can overcome the metabolic instability of CUR, resulting in increased oral bioavailability.
Revised sentences (page 3, lines 137-139):
The MPA-CUR conjugate may minimize the side effect of MPA if it is effective at a low dose and can overcome metabolic instability of CUR, resulting in an improvement of oral bioavailability.
- Original sentence (page 6, lines 313-316):
According to a previous publication by Franklin and colleagues, the chemical modification of phthalane and ringside chains led to MPA resistance to rapid conversion using cultures of human colorectal carcinoma, HT29 as a model for drug metabolism assays.
Revised sentences (page 6, lines 316-319):
According to the metabolism study of MPA in the culture media of human colorectal carcinoma (HT29) by Franklin and colleagues, the chemical modification of phthalane and ringside chains could lead to MPA resistance to rapid conversion [35].
- Original sentence (page 8, lines 369-372):
The MPA-CUR conjugate was synthesized successfully albeit in low yield, which may be attributed to the purification step even though the purification step has been optimized in terms of the mobile and stationary phase.
Revised sentence (page 8, lines 372-374):
The MPA-CUR conjugate was synthesized successfully, albeit in low yield. It may be attributed to the purification step even though the purification step has been optimized in terms of the mobile and stationary phases.
- Original sentence (page 7, lines 378-380):
We attempted to use silica gel column chromatography with a combination of hexane and DCM as the mobile phase, but the system failed to separate MPA-CUR from CUR, resulting in the co-elution of MPA-CUR and CUR.
Revised sentence (page 7, lines 380-383):
We attempted to use silica gel column chromatography with a combination of hexane and DCM as the mobile phase. Unfortunately, the system failed to separate MPA-CUR from CUR, resulting in the co-elution of MPA-CUR and CUR.
Point#2: The first sentence in the abstract contains unclear part:“Curcumin (CUR) has been used as adjuvant therapy for therapeutic application in the treatment of psoriasis by several mechanisms”.
Several mechanisms of treatment (methods?) or several mechanisms of curcumin signaling in cells? This sentence should be re-phrased.
Response:
Thank you for your comments. We have edited the first sentence of the abstract in the revised manuscript as shown below.
Original sentence:
Curcumin (CUR) has been used as adjuvant therapy for therapeutic application in the treatment of psoriasis by several mechanisms.
Revised sentence:
Curcumin (CUR) has been used as adjuvant therapy for therapeutic application in the treatment of psoriasis through several mechanisms of action.
Point#3: Line 48: “Natural products have shown an attractive bioactive source in drug discovery due to their unique structural diversity [7].” The structural diversity is not the main beneficial characteristics. The biological effect of the natural compounds is the reason why the agents are used for drug discovery. It is necessary to cite relevant references here.
Response:
Thank you very much for your corrections. We have edited this sentence in the revised manuscript as shown below. We have also added the relevant references here.
Original sentence (page 2, line 49):
Natural products have shown an attractive bioactive source in drug discovery due to their unique structural diversity [7].
Revised sentence (page 2, line 50):
Natural products have shown an attractive bioactive source in drug discovery due to their unique structural diversity, leading to many beneficial biological activities [7-9].
References:
- Dias, D.A., S. Urban, and U. Roessner, A historical overview of natural products in drug discovery. Metabolites, 2012. 2(2): p. 303-36.
- Lautie, E., et al., Unraveling Plant Natural Chemical Diversity for Drug Discovery Purposes. Front Pharmacol, 2020. 11: p. 397.
- Ekiert, H.M. and A. Szopa, Biological Activities of Natural Products. Molecules, 2020. 25(23).
Point#4: Fig. 6: western blots/cuts are too small for clear visualization; and should be increased in size. The size of the graphs can be decreased.
Response:
Thank you for your suggestion. We have resized Figure 6 as suggested in the revised manuscript.
Point#5: Study limitations should be described. Future directions (testing in vivo, which models ?…) should be indicated.
Response:
Thank you for raising this issue. We have added the limitations and future directions in the conclusion section of the revised manuscript (page 15, lines 678-682).
Added sentences in the conclusion section (page 15, lines 678-682).
It is worth noting that the CUR-MPA conjugate synthesized in this study will be considered as a novel chemical entity and will not be a product in the food list as curcumin [49]. To develop the conjugate as a potential drug candidate, further studies such as the investigation of physicochemical properties, biopharmaceutics classification, animal pharmacokinetics and in vivo antipsoriasis activities should be performed. Imiquimod (IMQ) induced wild-type mouse to mimic some features of psoriasis can be used as an animal model for an antipsoriasis assay for the conjugate.
References:
- Hewlings, S. J., & Kalman, D. S. (2017). Curcumin: A Review of Its Effects on Human Health. Foods (Basel, Switzerland), 6(10), 92. https://doi.org/10.3390/foods6100092
Point#6: Conclusions are too general. Avoid vacuous generalizations. Conclusions should indicate the findings (cytokines,proliferation-related effects etc.)
Response:
Thank you for your comments. We have revised the conclusion section of the revised manuscript as shown below.
Revised conclusion section (page 15, lines 667-682):
A mutual prodrug of MPA and CUR for the treatment of psoriasis has been synthesized using the Stegligh esterification between MPA and CUR. The MPA-CUR conjugate exhibited an improved antiproliferation effect against TNF-α induced HaCaT cells. The MPA-CUR conjugate was proposed as a mutual prodrug that the parent compounds showed synergistic action in the bioavailable fraction from Caco-2 cells. MPA-CUR inhibited inflammatory cytokines such as IL-6, IL-8 and IL-1β in TNF-α induced HaCaT cells through the attenuation of the MAPK signaling pathways including p38, ERK and JNK. These findings demonstrated the antiproliferative and anti-inflammatory effects of MPA-CUR that could be further developed for the treatment of psoriasis. It is worth noting that the CUR-MPA conjugate synthesized in this study will be considered as a novel chemical entity and will not be a product in the food list as curcumin [49]. To develop the conjugate as a potential drug candidate, further studies such as the investigation of physicochemical properties, biopharmaceutics classification, animal pharmacokinetics and in vivo antipsoriasis activities should be performed. Imiquimod (IMQ) induced wild-type mouse to mimic some features of psoriasis can be used as an animal model for an antipsoriasis assay for the conjugate.
